# EVOLUTIONARY POLICY OPTIMIZATION

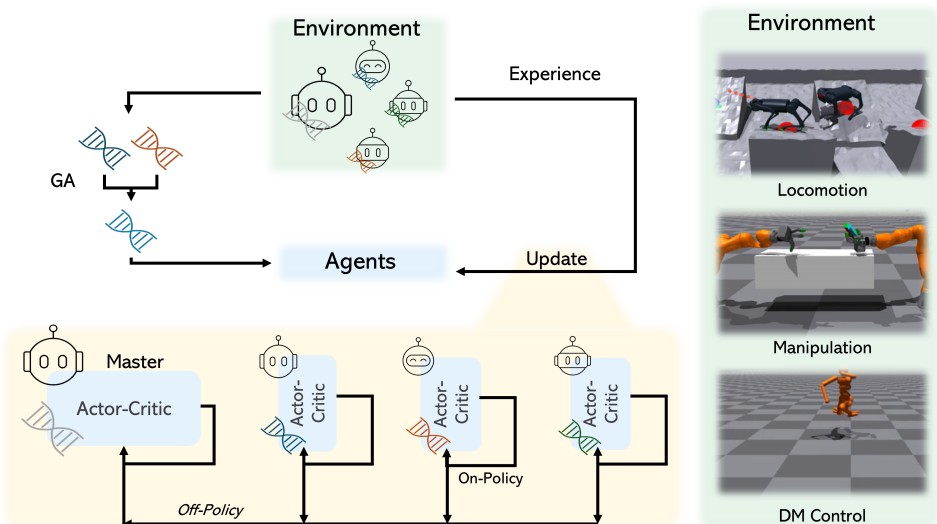

Figure 1: Evolutionary Policy Optimization (EPO) integrates genetic algorithms with policy gradients. A population of agents, represented by latent genes and sharing one actor-critic network, interacts with the environment. A master agent is trained on aggregated experiences from all agents to improve efficiency and stability.

## ABSTRACT

On-policy reinforcement learning (RL) algorithms are widely used for their strong asymptotic performance and training stability, but they struggle to scale with larger batch sizes, as additional parallel environments yield redundant data due to limited policy-induced diversity. In contrast, Evolutionary Algorithms (EAs) scale naturally and encourage exploration via randomized population-based search, but are often sample-inefficient. We propose Evolutionary Policy Optimization (EPO), a hybrid algorithm that combines the scalability and diversity of EAs with the performance and stability of policy gradients. EPO maintains a population of agents conditioned on latent variables, shares actor-critic network parameters for coherence and memory efficiency, and aggregates diverse experiences into a master agent. Across tasks in dexterous manipulation, legged locomotion, and classic control, EPO outperforms state-of-the-art baselines in sample efficiency, asymptotic performance, and scalability. For visualizations of the learned policies, please visit: https://sites.google.com/view/epo-rl.

## 1 INTRODUCTION

Reinforcement learning (RL) has become a powerful paradigm for training autonomous decision-making agents across a wide range of domains, including games Silver et al. (2016); Berner et al. (2019); Vinyals et al. (2019); Mnih (2016), robotics Kalashnikov et al. (2018); Akkaya et al. (2019); Kumar et al. (2021); Hwangbo et al. (2019), and large-language-model alignment Ouyang et al. (2022); Guo et al. (2025); Team et al. (2023). Like all trial-and-error methods, RL is inherently sample-inefficient, and the problem is particularly acute for *on-policy* algorithms, which discard past experience after each update.

Despite this drawback, on-policy model-free methods—commonly referred to as *policy gradients* Schulman et al. (2017); Mnih (2016)—are widely adopted in real-world applications Silver et al. (2016); Berner et al. (2019); Kalashnikov et al. (2018); Akkaya et al. (2019); Ouyang et al. (2022). Their popularity stems from their ability to achieve higher *asymptotic* returns and *stable* performance, making them the method of choice in data-rich settings such as simulations and game engines Makoviychuk et al., where large batches are easy to generate. Unfortunately, recent works Singla et al. (2024); Petrenko et al. (2023) show that policy-gradient algorithms do *not scale* well with larger batch sizes: because data are collected from the current policy, adding more parallel environments does not guarantee greater diversity. Instead, the data distribution quickly converges, so simply running more environments on additional GPUs yields highly correlated trajectories rather than useful variety.

Evolutionary algorithms (EAs) contribute the complementary strength of explicit *diversity* through random perturbations, making them natural partners for RL. Their combination—evolutionary RL (EvoRL) Moriarty et al. (1999); Salimans et al. (2017)—maintains a population of policies and refines them with evolutionary operators. Yet EvoRL has not been widely adopted. Classic EvoRL relies on gradient-free evolutionary strategies (ES) Khadka & Tumer (2018); Chen et al. (2019); Zheng & Cheng (2023), whose simplest variants reduce to random search Mania et al. (2018): they parallelize well but remain extremely sample-inefficient. Methods such as CEM-RL Pourchot & Sigaud (2019) introduce off-policy updates to improve efficiency but degrade asymptotic performance owing to distribution shift. Population-Based Training (PBT) Petrenko et al. (2023); Jaderberg et al. (2017) ensembles multiple on-policy agents, yet each policy evolves in isolation, wasting the opportunity to amortize experience across the population.

Can we design an algorithm that (i) scales with the number of parallel environments, (ii) learns efficiently from shared experience, and (iii) preserves the high asymptotic returns and stability of on-policy methods? We answer in the affirmative. We introduce **Evolutionary Policy Optimization (EPO)**, a novel policy-gradient algorithm that merges the strengths of EA and policy gradients: EA supplies diverse experience during massively parallel training, and experience from all agents is amortized to train a master agent via the split-and-aggregate policy-gradient (SAPG) formulation Singla et al. (2024), which performs off-policy updates with importance sampling to fold follower data into the master's on-policy set.

EPO maintains a population of agents alongside a master agent, all sharing the same network weights (analogous to gene-expression rules). This design prevents parameter bloat as the population scales, enabling efficient training and storage of large networks. Each agent is conditioned on a unique latent variable ("gene"), ensuring behavioral diversity while preserving coherence. Periodically, Darwinian selection Darwin (2023) is applied: low performers are removed, and elite agents are retained. These elites undergo crossover and mutation, injecting controlled diversity without excessive divergence. All agents are updated synchronously with policy gradients using shared reward signals. In every iteration, EPO aggregates experience from all agents into the master via SAPG updates Singla et al. (2024), enabling the master to learn from diverse, high-quality data and fully exploit massive parallelism.

We evaluate EPO on several challenging and realistic environments spanning manipulation (Isaac Gym Allegro-Kuka Makoviychuk et al.), locomotion (ANYmal Walking Hutter et al. (2016), Unitree A1 Parkour Cheng et al. (2024); Zhuang et al. (2023)), and classic control benchmarks (DeepMind Control Suite Tassa et al. (2018)). Across this suite, EPO significantly outperforms prior state-of-the-art RL methods. Moreover, our scaling-law analysis shows that EPO effectively scales with increased computational resources, making it a promising approach for large-scale RL training.

## 2 RELATED WORK

**On-Policy Reinforcement Learning** On-policy approaches Sutton et al. (1999) update the policy using data collected by the current policy, ensuring that the training data remains closely aligned with the policy's behavior. This strategy often leads to higher asymptotic rewards and stable performance, but at the cost of sample inefficiency, as past experiences from older policies are discarded. Despite this drawback, on-policy methods are the preferred choice in settings where data is abundant, such as simulations and game engines, where large training batches can be efficiently generated. In practice, on-policy RL has proven highly effective for training autonomous decision-

making agents in games Silver et al. (2016); Berner et al. (2019); Vinyals et al. (2019); Mnih (2016), robotics Kalashnikov et al. (2018); Akkaya et al. (2019); Kumar et al. (2021); Hwangbo et al. (2019), and more recently, in developing reasoning capabilities in large language models (LLMs) Ouyang et al. (2022); Guo et al. (2025); Team et al. (2023). Two widely used algorithms in this category are Trust Region Policy Optimization (TRPO) Schulman et al. (2015) and Proximal Policy Optimization (PPO) Schulman et al. (2017), both designed to stabilize learning by constraining the magnitude of policy updates.

**Off-Policy Reinforcement Learning** Unlike on-policy algorithms, off-policy methods allow for policy updates using data collected from different or older policies, enabling more efficient reuse of past experiences. Classic approaches such as Q-learning and its deep variant Deep Q-Network (DQN) Mnih (2013) rely on a replay buffer to store state-action transitions for more stable learning. More recent continuous control algorithms, such as Deep Deterministic Policy Gradient (DDPG) Lillicrap (2015); Silver et al. (2014), Twin Delayed DDPG (TD3) Fujimoto et al. (2018), and Soft Actor-Critic (SAC) Haarnoja et al. (2018), extend these ideas within an actor-critic framework, further improving stability and performance. While off-policy methods are typically more sample-efficient than on-policy methods, they often suffer from distribution shift and overestimation of action values, which can hinder stable learning. To effectively amortize experience across all policies in a population while ensuring stable updates, we adopt the Split and Aggregate Policy Gradient (SAPG) formulation Singla et al. (2024), which performs off-policy updates via importance sampling, enabling the aggregation of experience from individual follower policies into the on-policy data of a master policy.

**Evolutionary Reinforcement Learning** Another related line of work is Evolutionary Reinforcement Learning (EvoRL) Moriarty et al. (1999), which integrates population-based search with policy optimization to explore a broad range of candidate solutions. Historically, evolutionary computation (EC) has been employed to optimize neural network weights Tang et al. (2020); Choromanski et al. (2018; 2019), architectures Gaier & Ha (2019), and hyperparameters Jaderberg et al. (2017); Snoek et al. (2012); Vincent et al. (2024). For instance, OpenAI ES Chrabaszcz et al. (2018) applied a streamlined evolution strategy to Atari games. More recently, researchers have integrated gradient-based updates alongside evolutionary operations (e.g., mutation and crossover), improving scalability and leveraging the representational power of deep neural networks Pourchot & Sigaud (2019); Khadka et al. (2019). For example, Khadka & Tumer (2018) extends Deep Deterministic Policy Gradient (DDPG) by introducing evolutionary operations on a population of policies, combining the exploratory benefits of evolution with gradient-based optimization for more robust learning. However, these methods remain vulnerable to distribution shift, which undermines their asymptotic performance. In this work, we propose to combine the strengths of evolutionary search with on-policy updates, enabling higher asymptotic rewards while maintaining high sample efficiency.

## 3 Preliminaries

In this paper, we propose Evolutionary Policy Optimization (EPO), a novel reinforcement learning algorithm built on the well-established on-policy RL method Proximal Policy Optimization (PPO) and Genetic Algorithm (GA). Below, we provide a brief overview of these foundational methods.

**Reinforcement Learning** A standard reinforcement learning setting is formalized as a Markov Decision Process (MDP) and consists of an agent interacting with an environment $E$ over a number of discrete time steps. At each time step $t$, the agent receives a state $s_t$ and maps it to an action $a_t$ using its policy $\pi_\theta$. The agent receives a scalar reward $r_t$ and moves to the next state $s_{t+1}$. The process continues until the agent reaches a terminal state marking the end of an episode. The return $R_t = \sum_{k=0}^{\infty} \gamma^k r_{t+k}$ is the total accumulated return from time step $t$ with discount factor $\gamma \in (0, 1]$. The goal of the agent is to maximize the expected return.

**PPO** Proximal Policy Optimization is widely used in reinforcement learning applications. It is an actor-critic based policy gradient algorithm, where the key idea underlying policy gradients is to increase the probabilities of actions that lead to higher return and decrease the probabilities of actions that lead to lower return, iteratively refining the policy to achieve optimal performance.

Let $\pi_\theta$ denote a policy with parameters $\theta$, and $J(\pi_\theta)$ denote the expected finite-horizon undiscounted return of the policy. $J(\pi_\theta)$ is updated by:

$$\nabla_\theta J(\pi_\theta) = \mathop{\mathbb{E}}_{\substack{s \sim d^{\pi_\theta} \\ a \sim \pi_\theta(\cdot|s)}} \left[ \nabla_\theta \log \pi_\theta(a \mid s) A^{\pi_\theta}(s, a) \right], \tag{1}$$

where $A^{\pi_\theta}(s, a)$ is an advantage function that estimates the contribution of the transition to the gradient. A common choice is $A^{\pi_\theta}(s, a) = Q^{\pi_\theta}(s, a) - V^{\pi_\theta}(s)$, where $Q^{\pi_\theta}(s, a)$ and $V^{\pi_\theta}(s)$ are the estimated action-value and value functions, respectively. This form of update is termed an actor-critic update. Since we want the gradient of the error with respect to the current policy, only data from the current policy (on-policy) can be utilized. But these updates can be unstable because gradient estimates are high variance and the loss landscape is complex. An update step that is too large can degrade policy performance. Proximal Policy Optimization (PPO) modifies Eq. 1 to restrict updates to remain within an approximate *trust region* where improvement is guaranteed:

$$L_{on}(\pi_\theta) = \mathop{\mathbb{E}}_{\pi_{\text{old}}} \left[ \min \left( r_t(\pi_\theta) A_t^{\pi_{old}}, \text{clip}\left(r_t(\pi_\theta), 1 - \epsilon, 1 + \epsilon\right) A_t^{\pi_{old}} \right) \right], \tag{2}$$

where $r_t(\theta) = \frac{\pi_\theta(a_t|s_t)}{\pi_{\text{old}}(a_t|s_t)}$, $\epsilon$ is a clipping hyperparameter, and $\pi_{\text{old}}$ is the policy collecting the on-policy data. The clipping operation ensures that the updated $\pi_\theta$ stays close to $\pi_{\text{old}}$. Empirically, given large numbers of samples, PPO achieves high performance and is stable and robust to hyperparameters.

**Genetic Algorithm (GA)** is one of the most well-known Evolutionary Algorithms (EAs). It relies on three key operators: selection, mutation, and crossover. A core concept in genetic algorithms is the population, where each individual represents a potential solution evaluated using a fitness function. The classic genetic algorithm emphasizes crossover as the primary mechanism for exploration Hoffmeister & Bäck (1990) and is usually applied to deal with the problem of hyper-parameter tuning in reinforcement learning.

## 4 APPROACH

The core idea behind Evolutionary Policy Optimization (EPO) is to integrate the population-based exploration of Genetic Algorithms (GAs) with the powerful policy gradient. While our framework specifically combines GA with Proximal Policy Optimization (PPO), it can be extended to any on-policy RL algorithm utilizing an actor-critic architecture.

EPO (As shown in Figure 1) begins by initializing a population of agents, where each agent is represented by a unique latent embedding (gene), while all agents share a common actor-critic network (gene-expression rules). Among these agents, one is designated as the "master" agent. Each agent then interacts with the environment for a full episode, and its fitness value (defined differently for each task) is obtained during that episode. A selection operator then determines which agents survive to the next generation based on their fitness scores. The embeddings of the surviving agents undergo mutation and crossover to form the next generation, while a small subset of top-performing agents ("elites") is preserved unchanged to maintain stability.

To further enhance learning efficiency, we adopted a hybrid policy update strategy Singla et al. (2024). The master agent updates its parameters using both (i) on-policy data collected in the most recent episode and (ii) off-policy data sampled from the trajectories of the entire population. Meanwhile, each non-master agent is updated using only its own on-policy data. This hybrid learning process ensures both efficient exploration and sample reuse, making EPO well-suited for large-scale reinforcement learning. In the following sections, we describe each component of EPO in detail.

### 4.1 GENETIC ALGORITHM FOR AGENT SELECTION

The design philosophy of our algorithm is twofold: (1) to generate diverse yet bounded data for training the master agent and (2) to ensure that non-master agents are updated efficiently, enabling them to produce useful yet diverse data. Drawing inspiration from biological evolution, we propose

initializing a population of agents ($\{\pi_k\}_{k=1}^K$), where each agent is associated with a unique latent embedding ($\{\phi_k \in \mathcal{R}^N\}_{k=1}^K$). All agents share a common actor-critic network, parameterized by $\theta$ and $\psi$, respectively. The actor is conditioned on both the current state and its latent embedding to generate actions, while the value function is similarly conditioned to estimate value functions. Prior works have also leveraged latent-conditioned policies to encourage diverse behaviors Ebert et al. (2021); Eysenbach et al. (2018). Among these agents, one is designated as the master agent—for simplicity, we define the master agent as $\pi_1(\theta, \psi, \phi_1)$, though the general framework allows for alternative assignments.

To update non-master agents, we employ a genetic algorithm (GA) that modifies only the latent embeddings while keeping network parameters fixed. Each agent interacts with the environment for a full episode, and gets its fitness values. A **selection** operator determines which agents survive to the next generation, with survival probability proportional to their relative fitness scores (*i.e.*, cumulative reward, success rate, etc.). Specifically, we retain the top $x$ ranked non-master agents ($x \in [2, K]$) as elites, forming a set $X$, which is preserved unchanged and carried over to the next generation set $Y$. We then randomly select two agents from $X$ to undergo crossover and mutation. While multiple crossover methods exist (e.g., k-point crossover, uniform crossover), we use a simple averaging method for efficiency. Given two selected latent embeddings, $\phi_i, \phi_j$, where $\pi_i(\theta, \psi, \phi_i), \pi_j(\theta, \psi, \phi_j) \in X$, the **crossover** result is computed as: $\phi' = 1/2 \times (\phi_i + \phi_j)$. Following crossover, a **mutation** step is applied by adding Gaussian noise: $\phi'' = \phi' + \epsilon, \epsilon \sim \mathcal{N}(0, \sigma^2)$. The new agent $\pi(\theta, \psi, \phi'')$ is then added to $Y$ for the next generation.

It is important to note that for more complex tasks, such as dexterous manipulation, policies require longer training before they exhibit meaningful differences in performance. In such cases (*e.g.*, when all rewards are close to zero), performing genetic updates too early may be ineffective. To address this, we execute selection, crossover, and mutation only when the agents' performance differences become sufficiently significant. Specifically, the genetic algorithm is applied only when the difference between fitness scores exceeds a certain proportion of the median fitness score, ensuring that evolutionary updates occur only when meaningful behavioral differentiation has emerged.

---

**Algorithm 1: Evolutionary Policy Optimization**

---

1: Initialize $K$ agents $\{\pi_i\}_{i=1}^K \leftarrow (\{\phi_i\}_{i=1}^K, \theta, \psi)$
2: Initialize $K$ empty cyclic replay buffers $\{S\}_{i=1}^K$
3: Initialize $N$ environments $\{E\}_{i=1}^N$
4: **for** k = 1, ..., $K$ **do**
5:     $E^k = \{E\}_{i=\frac{(k-1)N}{K}}^{\frac{kN}{K}}$  ▷ Assign environments
6: **end for**
7: **for** iteration=1,2,... **do**
8:     **for** k = 2, ..., $K$ **do**
9:         $f_k \leftarrow Evaluate(E^k, \pi_k)$
10:     **end for**
11:     **if** Eq 3 **then**
12:         Rank population by fitness scores $f_k$
13:         Select top-$x$ as elites $X$, append to $Y$
14:         **while** $|Y| < K - 1$ **do**
15:             $\phi' \leftarrow$ Crossover($\phi_i, \phi_j$)
16:             $\phi'' \leftarrow$ Mutate($\phi'$)
17:             Append $\pi(\theta, \psi, \phi'')$ to $Y$
18:         **end while**
19:         $\{\pi_i\}_{i=2}^K \leftarrow Y$
20:     **end if**
21:     **for** k = 1, ..., $K$ **do**
22:         CollectData($E^k, \pi_k$)
23:     **end for**
24:     Sample $|S_1|$ transitions from $\cup_{k=2}^K S_k$ to get $S_1'$
25:     $L \leftarrow$ OffPolicyLoss($S_1'$)
26:     **for** k = 1, ..., $K$ **do**
27:         $L \leftarrow L+$ OnPolicyLoss($S_k$)
28:     **end for**
29:     Update $\theta, \psi, \{\phi_k\}_{k=1}^K$
30: **end for**
31: Return $\theta, \psi, \phi_1$

---

$$\max\{f_k\}_{k=2}^K - \min\{f_k\}_{k=2}^K > \gamma \cdot \text{median}\{f_k\}_{k=2}^K \tag{3}$$

, where $f_k$ represent the fitness score of agent $\pi_k$. A simpler alternative is to apply selection, crossover, and mutation at fixed intervals—e.g., every set number of environment steps.

All parameters ($\theta, \psi, \{\phi_k\}_{k=1}^K$) are updated using the following hybrid policy optimization process.

## 4.2 HYBRID-POLICY OPTIMIZATION

Different agents generate diverse data through varied behaviors, which can be beneficial if the master agent can effectively learn from all available data. However, since data distributions collected by different agents may vary significantly, incorporating off-policy data requires careful correction.

Following Meng et al. (2023); Singla et al. (2024), we apply importance sampling to reweight updates from different policies. Specifically, we sample $|S_1|$ transitions from $\cup_{k=2}^{K} S_k$ collected by agents $\{\pi_k\}_{k=2}^{K}$ to get $S_1'$. To update the master agent $\pi_1$ using $S_1'$, we define the off-policy loss as:

$$L_{\text{off}}(\pi_1, S_1') = \frac{1}{|S_1'|} \underset{(s,a) \sim S_1'}{\mathbb{E}} [\min(r_{\pi_1}(s,a)A^{\pi_1,\text{old}}(s,a), \quad \text{clip}(r_{\pi_1}(s,a), \mu(1-\epsilon), \mu(1+\epsilon))A^{\pi_1,\text{old}}(s,a))] \tag{4}$$

where $r_{\pi_1}(s,a)$ is the importance sampling ratio, given by $r_{\pi_1}(s,a) = \frac{\pi_1(s,a)}{\pi_k(s,a)}$ and $\mu$ is an off-policy correction term, given by $\mu = \frac{\pi_{1,old}(s,a)}{\pi_k(s,a)}$. The final policy loss combines this off-policy loss with the on-policy loss given by Equation 2:

$$L(\pi_1) = L_{on}(\pi_1) + \lambda \cdot L_{off}(\pi_1, S_1') \tag{5}$$

**Critic Update with Hybrid Data**    The target value for the critic is computed using $n$-step returns (where $n = 3$) for on-policy data:

$$\hat{V}_{\text{on},\pi_k}^{\text{target}}(s_t) = \sum_{m=t}^{t+2} \gamma^{m-t} r_m + \gamma^3 V_{\pi_k,\text{old}}(s_{t+3}) \tag{6}$$

The critic loss for on-policy data is then:

$$L_{\text{on}}^{\text{critic}}(\pi_k) = \underset{(s,a) \sim \pi_k}{\mathbb{E}} \left[ \left( V_{\pi_k}(s) - \hat{V}_{\text{on},\pi_k}^{\text{target}}(s) \right)^2 \right] \tag{7}$$

For off-policy data, however, multi-step returns are not directly applicable. Instead, we approximate the 1-step return:

$$\hat{V}_{\text{off},\pi_1}^{\text{target}}(s_t') = r_t + \gamma V_{\pi_1,\text{old}}(s_{t+1}') \tag{8}$$

The critic loss for off-policy data is then:

$$L_{\text{off}}^{\text{critic}}(\pi_1, S_1') = \frac{1}{|S_1'|} \underset{(s,a) \sim S_1'}{\mathbb{E}} \left[ \left( V_{\pi_1}(s) - \hat{V}_{\text{off},\pi_1}^{\text{target}}(s) \right)^2 \right] \tag{9}$$

The final critic loss is a weighted combination of the on-policy and off-policy losses:

$$L^{\text{critic}}(\pi_1) = L_{\text{on}}^{\text{critic}}(\pi_1) + \lambda \cdot L_{\text{off}}^{\text{critic}}(\pi_1, S_1') \tag{10}$$

**Updates for Non-Master Agents**    All agents, except for the master agent $\pi_1$, are updated exclusively using on-policy losses, as defined in Equations 2 and 6. This ensures that while the master agent leverages additional experience from off-policy data, the remaining agents maintain stability by learning from their own trajectories, thereby preserving diversity within the population.

## 5    EXPERIMENTS

We evaluate our method on eight challenging and realistic environments, including manipulation Petrenko et al. (2023), locomotion Hutter et al. (2016); Cheng et al. (2024), and classic control benchmarks Tassa et al. (2018), comparing it against state-of-the-art (SOTA) approaches. Additionally, we conduct ablation studies to assess the contribution of each component. To demonstrate scalability, we analyze our method's performance with increasing computational resources. Finally, we evaluated computational cost versus performance gain as EPO scales with agent numbers (Supplementary Material). Each aspect is discussed in detail.

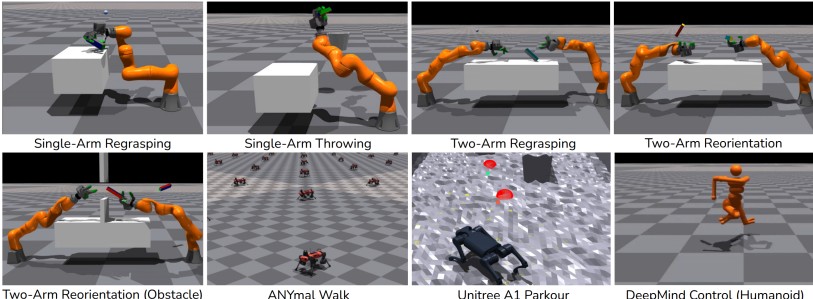

Figure 2: We evaluate our algorithm on eight challenging environments that span a diverse set of tasks, including manipulation Petrenko et al. (2023), locomotion Cheng et al. (2024), and classic control benchmarks Tassa et al. (2018)

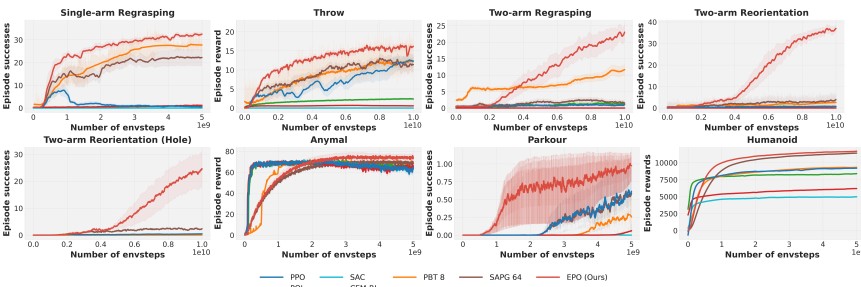

Figure 3: Performance curves of EPO compared to SAC, PQL, PPO, SAPG 64, PBT 8, and CEM-RL baselines. We plot best-performing agent numbers for SAPG (64) and PBT (8). EPO shows superior sample efficiency and higher asymptotic performance, particularly on challenging tasks.

## 5.1 EXPERIMENTAL SETUP

**Tasks** We evaluate our algorithm across eight challenging environments covering a diverse range of tasks, including manipulation Petrenko et al. (2023)—Single-Arm Regrasping (M:SG), Single-Arm Throwing (M:ST), Two-Arm Regrasping (M:TG), Two-Arm Reorientation (M:TR), and Two-Arm Reorientation with Obstacle (M:TO); locomotion—ANYmal Walking (L:A)Hutter et al. (2016) and Unitree A1 Parkour (L:U)Cheng et al. (2024); and classic control—Humanoid from the DeepMind Control Suite (D)Tassa et al. (2018). A visual overview of these tasks is shown in Figure2; additional details are provided in the Appendix.

Following prior works Petrenko et al. (2023); Singla et al. (2024), we use the number of successes per episode as the performance metric for manipulation tasks. For L:A and D, we report the cumulative rewards as defined in their original benchmarks. For L:U, we use the average terrain difficulty level achieved by all A1 agents during an episode, which serves as a proxy for success rate.

**Baselines** We evaluate our approach against state-of-the-art RL methods tailored for large-scale training. The comparisons include off-policy algorithms—SAC Haarnoja et al. (2018) and Parallel Q-Learning Li et al. (2023); on-policy methods—PPO Schulman et al. (2017); hybrid-policy methods—SAPG Singla et al. (2024); and population-based EvoRL methods—PBT Petrenko et al. (2023) and CEM-RL Pourchot & Sigaud (2019). We use the original implementations of each method without introducing algorithm-level modifications. Please refer to the supplementary material for additional details.

For a fair comparison, all algorithms are trained using the same number of environments ($N = 24{,}576$) and identical network architectures. For methods incorporating a population-based component (SAPG, PBT, CEM-RL), we standardize the population size to 64. Since the impact of the number of agents on final performance can vary, we also report results using each method's original agent configuration for completeness (e.g., PBT: 8 agents; SAPG: 6 agents) in the Table 1. Each experiment is run with 5 different random seeds, and we report the mean and standard error in the plots. Further ablation studies on the effect of population size are presented in Section 5.3.

| Task | SAC | PQL | PPO | SAPG 64 | SAPG 6 | PBT 64 | PBT 8 | CEM-RL | EPO (Ours) |
|------|-----|-----|-----|---------|--------|--------|-------|--------|------------|
| M:SG | $0.1 \pm 0.00$ | $3.1 \pm 0.31$ | $0.1 \pm 0.11$ | $22.4 \pm 7.60$ | $31.6 \pm 2.07$ | $6.8 \pm 5.01$ | $27.7 \pm 1.24$ | $1.7 \pm 0.95$ | $\mathbf{32.1 \pm 1.21}$ |
| M:ST | $0.0$ | $2.5 \pm 0.43$ | $12.4 \pm 3.11$ | $10.8 \pm 1.70$ | $11.5 \pm 4.15$ | $2.2 \pm 1.54$ | $12.1 \pm 1.22$ | $0.0$ | $\mathbf{16.4 \pm 0.99}$ |
| M:TG | $0.0$ | $1.7 \pm 0.30$ | $0.3 \pm 0.15$ | $1.4 \pm 1.04$ | $4.1 \pm 2.04$ | $3.1 \pm 1.24$ | $11.6 \pm 3.23$ | $0.0$ | $\mathbf{24.1 \pm 2.55}$ |
| M:TR | $0.0$ | $0.5 \pm 0.22$ | $0.7 \pm 0.45$ | $4.4 \pm 4.17$ | $3.9 \pm 3.24$ | $0.0$ | $3.0 \pm 1.34$ | $0.0$ | $\mathbf{37.0 \pm 3.16}$ |
| M:TO | $0.0$ | $0.0$ | $0.5 \pm 0.43$ | $2.6 \pm 0.76$ | $2.4 \pm 0.52$ | $0.0$ | $0.0$ | $0.0$ | $\mathbf{24.7 \pm 7.06}$ |
| L:A | $64.2 \pm 6.60$ | $65.1 \pm 3.2$ | $67.7 \pm 1.71$ | $69.8 \pm 1.81$ | $68.9 \pm 1.22$ | $55.2 \pm 1.72$ | $64.7 \pm 6.16$ | $69.6 \pm 3.08$ | $\mathbf{74.0 \pm 6.65}$ |
| L:U | $0.0$ | $0.1 \pm 0.03$ | $0.6 \pm 0.32$ | $0.6 \pm 0.45$ | $0.6 \pm 0.45$ | $0.0$ | $0.3 \pm 0.31$ | $0.1 \pm 0.09$ | $\mathbf{1.0 \pm 0.45}$ |
| D | $4910.8 \pm 113.65$ | $8554.8 \pm 593.32$ | $9231.4 \pm 88.47$ | $11411.9 \pm 790.88$ | $11524.1 \pm 540.23$ | $7510.2 \pm 117.86$ | $9510.5 \pm 35.12$ | $7040.5 \pm 103.09$ | $\mathbf{11686.5 \pm 773.7}$ |

Table 1: Performance of EPO compared to SAC, PQL, PPO, SAPG, PBT, and CEM-RL across a diverse set of tasks. In manipulation tasks, PPO, SAPG, and PBT demonstrate learning capabilities in simple scenarios; however, only EPO achieves significant learning progress on complex tasks. For locomotion tasks, where PPO is widely regarded as a strong baseline and the default method, EPO still achieves superior performance by a large margin on the more challenging problems.

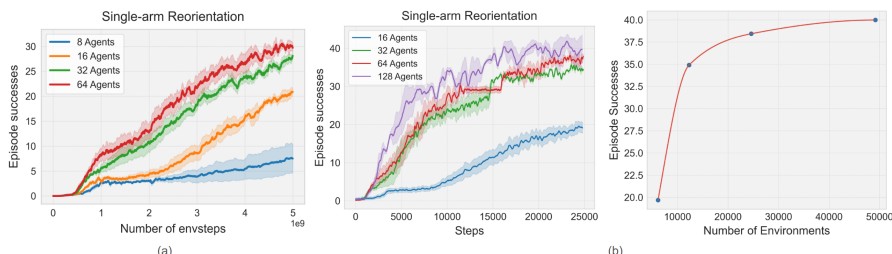

(a)            (b)

Figure 4: Ablation and scaling results for EPO. (a) Ablation of agent number ($K = 8, 16, 32, 64$) with a fixed total number of environments ($N = 24{,}576$). As the number of agents increases, EPO's performance improves. (b) Training and scaling curves with increasing environments. With a fixed number of environments per agent (384), total performance improves as environments scale.

## 5.2 RESULTS AND ANALYSIS

As shown in Figure 3 and Table 1, our algorithm consistently and significantly outperforms all baselines. For more visualization, please refer to https://sites.google.com/view/epo-rl.

In manipulation tasks, while methods like PPO, SAPG, and PBT perform reasonably on simpler tasks (e.g., single-arm regrasping and throwing), EPO still achieves higher success rates. For more complex tasks requiring two-arm coordination, baselines fail to learn meaningful behavior even after $1 \times 10^{10}$ steps, whereas EPO achieves strong performance. For example, EPO reaches 35.8 episode successes in two-arm reorientation using $1 \times 10^{10}$ steps, surpassing SAPG's 28.58 after $5 \times 10^{10}$ steps in the original SAPG paper Singla et al. (2024), highlighting EPO's superior sample efficiency. A similar trend holds in locomotion. On flat-ground ANYmal walking, baselines match EPO, but in more complex parkour tasks requiring rich exploration, EPO significantly outperforms them. In the DeepMind Control Suite (Humanoid), all methods perform reasonably, yet EPO still yields marginally higher returns.

Comparing SAPG 64 (5th column) with SAPG 6 (6th column) reveals that hybrid-policy-only updates do not benefit from large-scale simulation and may even degrade performance, as increasing the number of agents introduces a higher proportion of poorly performing policies. In contrast, comparing SAPG 64 (5th column) with EPO (last column) highlights the critical role of the genetic algorithm (Table 1), especially in more challenging tasks like two-arm manipulation. By filtering out underperforming agents and learning only from high-quality data, the GA ensures stable and effective training. This demonstrates that the genetic algorithm is not a minor enhancement, but a fundamental component of the hybrid-policy update.

Notably, EPO demonstrates a lower performance variance across different random seeds compared to baseline methods, indicating strong robustness to hyperparameter variations, including random seed initialization. This is especially significant given the well-known sensitivity of reinforcement learning algorithms to hyperparameter settings. EPO's robustness makes it a compelling choice for training RL models with minimal hyperparameter tuning, addressing a major challenge in deploying RL in real-world applications.

Compared to EvoRL baselines (PBT and CEM-RL), EPO demonstrates superior learning efficiency and asymptotic performance. While PBT performs well on simpler tasks, its efficiency decreases on

more complex ones. In PBT, each agent learns only from its own experience, and the data collected by eliminated agents is discarded rather than reused. This limitation becomes more pronounced in complex tasks that require significantly more environment steps for training. Consequently, PBT does not benefit from larger population sizes, as evidenced by the similar performance between PBT 64 and PBT 8. In contrast, CEM-RL relies heavily on off-policy updates, which becomes a limitation in difficult tasks where most agents perform poorly. This leads to the accumulation of low-quality data and ultimately degrades asymptotic performance across a wide range of tasks. EPO overcomes these limitations by employing a hybrid-policy update mechanism while retaining the genetic algorithm for agent selection, particularly in complex environments.

### 5.3 Ablations

In this ablation study, we investigate the impact of population size while keeping the total computational budget fixed, using the single-arm reorientation task—where the agent must grasp an object and reorient it to a target 6-DoF pose—as the testbed. Unless otherwise specified, all subsequent analyses in this section are conducted within this environment. Specifically, we fix the number of environments to $N = 24{,}576$ and vary the population size from $K = 8$ to $K = 64$ (Figure 4(a)). The results indicate that our algorithm benefits from a larger population. While this may seem intuitive, it is not a guaranteed outcome—larger populations introduce greater behavioral diversity, which can also increase the prevalence of poorly performing agents and result in noisier data. To address this, EPO applies Genetic Algorithms (GA) at the latent space level to periodically eliminate underperforming agents, ensuring that only high-quality data contributes to learning. Moreover, since all agents share a common actor-critic network, we maintain controlled diversity while avoiding unbounded policy divergence—an essential property for stable and scalable training.

We analyzed the impact of different crossover strategies beyond simple averaging, testing two alternatives: uniform crossover and fitness-weighted averaging. In uniform crossover, each element of the latent vector is randomly chosen from one parent with equal probability, while in fitness-weighted averaging, vectors are combined proportionally to fitness scores. Our method achieved a 36.7% success rate, compared to 32.6% for uniform crossover and 37.0% for fitness-weighted averaging, showing robustness to the crossover choice. For simplicity, we keep the averaging strategy.

### 5.4 Scaling Law

Here, we examine the scalability of EPO with increased computational resources. Specifically, we evaluate EPO's scalability with larger training batches generated by parallel simulations by fixing the number of environments per agent and increasing the number of agents from $K = 16$ to $K = 128$, thereby scaling the total number of environments from $N = 6{,}144$ to $N = 49{,}152$. It is worth noting that even at $K = 16$, the batch size of $N = 6{,}144$ is already significantly large. As shown in Figure 4(b), EPO continues to improve with increasing training data, even when the number of environments reaches $N = 49{,}152$, demonstrating that our algorithm overcomes the limitations of existing on-policy RL methods, which typically saturate beyond a certain batch size. This highlights EPO's strong potential for data-rich environments, such as simulations and game engines, where large training batches can be easily generated and leveraged effectively.

It is worth noting that our method focuses on scaling with training data, which is orthogonal to efforts that scale performance through larger neural network architectures. This makes it naturally compatible with existing regularization techniques designed for training large networks Nauman et al. (2024); Lee et al. (2024). Additionally, the shared actor-critic architecture enhances memory efficiency and inherently supports scaling to larger network sizes.

## 6 Conclusion

In conclusion, we present Evolutionary Policy Optimization (EPO), a novel policy gradient algorithm that integrates evolutionary algorithms with policy optimization. We demonstrate that EPO significantly outperforms state-of-the-art baselines across several challenging RL benchmarks while also scaling effectively with increased computational resources. Our approach enables large-scale RL training while maintaining high asymptotic performance, and we hope it inspires future research in advancing scalable reinforcement learning.

ETHICS STATEMENT

While EPO does not present direct ethical concerns, RL methods, particularly those that improve large-scale training, could have broader societal implications. Scalable RL has potential applications in autonomous systems, decision-making AI, and robotics, which may impact labor markets, automation policies, and safety regulations. It is crucial to ensure that future applications of such methods align with ethical AI principles, particularly regarding fairness, transparency, and robustness in high-stakes decision-making.

Additionally, as RL is increasingly applied to large-scale language models (LLMs) and automated reasoning systems, scalability improvements may accelerate advancements in AI alignment and human-AI interaction. However, misuse of reinforcement learning in generating persuasive AI or manipulative automated systems remains a potential risk. We encourage the community to apply EPO responsibly and consider ethical safeguards when deploying large-scale RL systems in real-world applications.

REPRODUCIBILITY STATEMENT

For reproducibility, we have provided an anonymous link to the downloadable source code. Please refer to https://sites.google.com/view/epo-rlfor more details.

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

## A  COMPUTATIONAL COST

We also evaluate the computational cost of EPO compared with other baselines that involve a population-based component (SAPG, PBT, CEM-RL). Since EPO requires calculating fitness score of all agents and determining whether iteration is necessary, it slightly increases the computational cost over PBT and SAPG. However, simply increasing the number of agents in EPO does not have a significant impact on memory usage or inference time. This is because the additional cost mainly comes from adding a small number of latent embeddings, which does not drastically affect the overall resource usage. All results are obtained on task M:SO with 24,576 environments and $5e9$ environment steps.

| Method | Speed (fps) | GPU Memory Usage (GB) | Performance |
|---|---|---|---|
| EPO (64 agents) | 6764 | $\sim 24$ GB | 29.8 |
| PBT (64 agents) | 8120 | $\sim 120$GB | 1.9 |
| SAPG (64 agents) | 6672 | $\sim 23$GB | 13.4 |
| CEM-RL (64 agents) | 9230 | $\sim 20$GB | 0.0 |
| EPO (8 agents) | 6432 | $\sim$21.4 GB | 8.4 |
| EPO (16 agents) | 6650 | $\sim$21.9 GB | 21.2 |
| EPO (32 agents) | 6685 | $\sim$22.9 GB | 28.1 |
| EPO (64 agents) | 6764 | $\sim$24GB | 29.8 |

## B  EXPERIMENT DETAILS

In this section, we present more details of our experiments.

### B.1  TASK AND EVALUATION DETAILS

- **Manipulation: Single-Arm Regrasping (M:SG)**: The agent must lift a cube from the table and hold it near a randomly generated target point (shown as a white sphere in Figure 2) within 30 steps. A success is defined as the center distance between the cube and the target point being within a predefined threshold.

- **Manipulation: Single-Arm Throwing (M:ST)**: The agent need pick up the cube and throw it to a bucket at a random position. A succcess is defined as if the cube is throwed into the bucket.

- **Manipulation: Single-Arm Reorientation (M:SO - used for ablations)**: The agent must pick up an object and reorient it to a specified 6-DoF pose including both position and orientation. A success is defined as both the positional and orientation differences between the object and the target pose being within a threshold.

- **Manipulation: Two-Arm Regrasping (M:TG)**: The goal is the same as in one-arm regrasping. However, the generated goal pose is far from the initial pose, requiring two Kuka arms to collaborate to transfer (*e.g.* throwing) the object to reach target poses in different regions.

- **Manipulation: Two-Arm Reorientation (M:TR)**: The goal is the same as in one-arm reorientation. However, the target pose is placed far from the initial pose, necessitating coordinated actions between the two Kuka arms.

- **Manipulation: Two-Arm Reorientation with Obstacle (M:TO)**: The task shares the same objective as two-arm reorientation, but with an obstacle in the center of the table. The agent must learn to pass the cube through the obstacle in order to successfully align it with the target orientation.

- **Locomotion: ANYmal (L:A)**: A quadruped robot is tasked with following velocity commands across flat terrain.

- **Locomotion: Unitree A1 Parkour (L:U)**: In this task, the robot dog needs to learn how to traverse complex terrains, such as hurdles, gaps, rough terrain, etc. The robot dog will

advance to the next, more difficult terrain only after it can successfully navigate the current one. Therefore, we consider the average terrain difficulty level that the robot dogs are on as the success rate. A value of 1 means that, on average, all robot dogs can complete the terrain at the first difficulty level.

- **DeepMind Control Suite: Humanoid (D)**: A 21-joint humanoid must run forward at a target velocity of 10 m/s without falling.

## B.2 HYPERPARAMETERS AND NETWORK

**Manipulation Tasks**   We employ a Gaussian policy in which the mean is generated by a single-layer LSTM with 768 hidden units. The input observation is first processed by an MLP with hidden layers of sizes 768, 512, and 256, using ELU activation, before being fed into the LSTM.

| Hyperparameter | Value |
|---|---|
| Discount factor | 0.99 |
| Learning rate | $1e - 4$ |
| KL threshold for LR update | 0.016 |
| Num Envs | $24, 576$ |
| Mini-batch size | $4 \cdot$ num_envs |
| Grad norm | 1.0 |
| Clipping factor | 0.1 |
| Critic coefficient | 4.0 |
| Horizon length | 16 |
| LSTM Sequence length | 16 |
| Bounds loss coefficient | 0.00001 |
| Mini Epochs | 2 |
| $\gamma$ in Eq.(3) | 0.3 |
| $\lambda$ in Eq.(5) | 1.0 |
| $x$ in Line 13 of Algo.1 | num_agents$(K) - 2$ |

**Locomotion Task**   We adopt a Gaussian policy in which the mean is produced by a MLP with hidden layer sizes of 512, 256, and 128, using ELU activation functions.

| Hyperparameter | Value |
|---|---|
| Discount factor | 0.99 |
| Learning rate | $1e - 4$ |
| KL threshold for LR update | 0.016 |
| Num Envs | $24, 576$ |
| Mini-batch size | $4 \cdot$ num_envs |
| Grad norm | 1.0 |
| Clipping factor | 0.1 |
| Critic coefficient | 4.0 |
| Horizon length | 16 |
| Bounds loss coefficient | 0.00001 |
| Mini Epochs | 2 |
| $\gamma$ in Eq.(3) | 0.5 |
| $\lambda$ in Eq.(5) | 1.0 |
| $x$ in Line 13 of Algo.1 | num_agents$(K) - 2$ |

## B.3 THE USE OF LARGE LANGUAGE MODELS (LLMS)

We use LLMs solely for polishing writing and checking grammar.

