# OpenReview forum: "Evolutionary Policy Optimization"
_ICLR.cc/2026/Conference — Submitted to ICLR 2026_

### Official Review · Reviewer_QE9Y · 2025-10-24

**Soundness:** 2
**Presentation:** 3
**Contribution:** 2
**Rating:** 4
**Confidence:** 3

**Summary:**

The paper proposes Evolutionary Policy Optimization (EPO), a hybrid reinforcement learning method combining Genetic Algorithms (GAs) and on-policy gradients (PPO) via Split-and-Aggregate Policy Gradients (SAPG). Agents share a single actor–critic network but differ through latent embeddings (“genes”), evolved by GA to maintain diversity. A master policy aggregates all agents’ experience through importance-weighted updates.
Experiments on 8 tasks (manipulation, locomotion, control) show that EPO outperforms PPO, SAPG, PBT, and CEM-RL, and appears to scale better with parallel environments.

**Strengths:**

- Clear motivation and solid integration of known techniques (PPO + GA + SAPG).
- Strong empirical gains, especially in complex manipulation tasks.
- Elegant, memory-efficient design using shared weights and latent conditioning.
- Results demonstrate robustness across seeds and domains.

**Weaknesses:**

See the questions mainly but here is summary;
- Limited novelty: Most components are borrowed from prior work (CEM-RL, SAPG, DIAYN).
- Scaling evidence insufficient: §5.4’s scaling test (up to 49 k envs) lacks baseline comparisons and deeper analysis.
- Ablation mismatch: §5.3 fixes total environments, testing population diversity rather than true scalability.

**Questions:**

### General questions
- What is the dimension of the latent embeddings (genes)? How sensitive is performance to this size?
- The paper notes prior work on latent-conditioned policies for diversity / skills (e.g., Ebert et al., 2021; Eysenbach et al., 2018). Could the authors clarify how EPO’s use of latent embeddings (“genes”) differs from these earlier approaches like DIAYN, which also condition policies on latent variables to induce behavioural diversity? Specifically, what new insight does your GA-based evolution add?

### Figure 3 and Table 1
In the experiments, the total number of environments is fixed at 24 576 and split evenly among agents (e.g., SAPG 6 → 4096 envs / agent, SAPG 64 and EPO 64 → 384 envs / agent). Could the authors clarify whether EPO also follows this same split? If so, how do they ensure that the observed performance gains stem from algorithmic scalability rather than simply from the diversity induced by the latent-conditioned population under reduced per-agent rollouts?

### $5.3
The total number of environments (24 576) is fixed while varying population size (K = 8–64), meaning only the split per agent changes. Since EPO claims to overcome the scalability cap when increasing total parallel environments, why is N held constant here? Doesn’t this setup test population diversity rather than true scalability with additional rollouts?

### $5.4
Since EPO’s main claim is improved scalability with increasing parallel environments, §5.4’s limited scaling experiment (up to 49k envs, no detailed comparison) feels insufficient. Could the authors expand the scaling analysis to better support this central claim?

---

### Official Review · Reviewer_UaU3 · 2025-10-28

**Soundness:** 2
**Presentation:** 3
**Contribution:** 2
**Rating:** 4
**Confidence:** 5

**Summary:**

The paper proposes Evolutionary Policy Optimization (EPO): it maintains a population of agents, each controlled by a latent embedding, while all agents share a common actor–critic network. A designated master agent aggregates experience from the entire population and performs off-policy updates; the remaining agents conduct on-policy PPO updates. Evolutionary operations (selection, crossover, mutation) are applied only to the latent embeddings, not to the network parameters. Experiments span manipulation, legged/humanoid locomotion, and classic control tasks, showing that EPO outperforms several baselines in sample efficiency, final performance, and scalability with parallel environments.

**Strengths:**

- The writing is clear and the method is easy to follow.
- The approach retains performance gains when increasing population size and parallel environment count, aligning with the common practice of scaling RL by boosting throughput.

**Weaknesses:**

- The paper suffers from insufficient baseline coverage and ambiguous positioning: there exist many recent hybrid EA-RL[1-4] methods (e.g., PDERL[5], PGPS[7], CSPS[6], ERL-Re²[8], EvoRainbow[9]). Comparing to older methods such as PBT and CEM-RL (pre-2018) does not convincingly demonstrate effectiveness. The lack of comprehensive literature mapping undermines readers’ ability to assess the core contribution.

- Most of the design components (on-policy/off-policy mix, population+latent embeddings) have appeared in previous work (e.g., CSPS used on/off-policy combination). The authors should clearly articulate how their method differs from CSPS and other ERL methods, and highlight the novel aspects.

- The experiments lack comparison on established benchmarks with other advanced hybrid algorithms (e.g., MetaWorld, MuJoCo, DMC).

- Some fine-grained implementation details may have large effects on results (e.g., using 3-step TD returns while comparison methods may not). The authors should provide ablations to rule out that performance gains come from hyper-parameter tuning.

- The authors should explicitly distinguish between two types of scaling: (A) increasing parallel throughput (N ↑) and (B) increasing model parameters.

- The evolutionary trigger using range/median is heuristic; a sensitivity scan over γ and a comparison with fixed-interval trigger or trend-based trigge are needed, especially in sparse reward environments.

- Missing technical details: recommend consolidating list of elite count, crossover method, mutation noise, latent dimension & injection location (actor/critic symmetric or not), replay capacity/sampling strategy, IS clipping bounds, etc.; also provide scripts and random seeds.

---
[1]. Bridging Evolutionary Algorithms and Reinforcement Learning: A Comprehensive Survey on Hybrid Algorithms. IEEE TEVC 2024.

[2]. Reinforcement learning versus evolutionary computation: A survey on hybrid algorithms. Swarm Evol. Comput. 2019.

[3]. Evolutionary reinforcement learning: A survey.

[4]. A survey on evolutionary reinforcement learning algorithms. Neurocomputing, 2023.

[5]. Proximal distilled evolutionary reinforcement learning, AAAI 2020.

[6]. Cooperative heterogeneous deep reinforcement learning, NeurIPS 2020.

[7]. PGPS: Coupling policy gradient with population-based search.

[8]. ERL-Re^2: Efficient evolutionary reinforcement learning with shared state representation and individual policy representation. ICLR 2023.

[9]. EvoRainbow: Combining improvements in evolutionary reinforcement learning for policy search. ICML 2024.

**Questions:**

- I noticed that many baselines in the paper were run for 1 billion steps, for off-policy algorithms, how long did this actually take in terms of wall-clock time and compute resources?
- Under fixed compute budget and fixed step count, please provide comparisons for returns with \(n \in \{1,3,5\}\) and \(\lambda \in \{0.8, 0.9, 0.95\}\); what are the differences in performance?

---

### Official Review · Reviewer_7CXx · 2025-10-30

**Soundness:** 2
**Presentation:** 2
**Contribution:** 2
**Rating:** 2
**Confidence:** 4

**Summary:**

The paper proposes Evolutionary Policy Optimization (EPO): a shared actor–critic approach with agent diversity induced via a latent “gene” per agent and applying evolutionary operators (selection/crossover/mutation) in the latent space. A master policy is trained with a split-and-aggregate (SAPG) approach with importance-weighted off-policy correction that uses data from follower agents. Results on challenging continuous-control tasks show strong returns and good scaling versus PPO/SAC/PBT/CEM-RL.

**Strengths:**

- The idea of maintaining a shared actor-critic with a latent "gene" per agent is interesting to reduce memory efficiency. The increased "coherence" between the agents would come at the cost of diversity; however, it's an interesting and meaningful design knob.
- The empirical results are on challenging high-dimensional tasks and show a strong task performance.
- The idea of combining population-based exploration with standard policy gradient is interesting and well-motivated by prior work.
- Scaling with a larger number of agents shows a clear trend.

**Weaknesses:**

- The paper's key contributions are unclear from the writing and experiments. The individual components like latent conditioning, shared weights, and evolutionary algorithms are known, so it's not clear where the novelty lies, and what isolated novel contributions drive gains. The paper does not explicitly mention the contributions, and there are no ablations of individual components of the supposed contributions.
- Section 4.2's off-policy loss needs an algorithmic or theoretical justification that it doesn't break any assumptions of the underlying algorithms and would preserve its convergence properties.
- It's great to see gains with parallelism and scaling; however, some experiments in a limited data regime would also clarify the limitations and capabilities of this method.
- The differential use of multi-step returns is also not justified empirically. It's tough to understand what the real reasons are that drive the gains.
- "EPO demonstrates a lower performance variance across different random seeds compared to baseline methods" - from the figure, this claim does not seem justified. Are there any quantitative results that the variance over seeds and hyperparameter variations is necessarily reduced with EPO? And what is the motivation for such variations to be reduced?

**Questions:**

- Could the authors provide clear definitions of their contributions over prior work and perform surgical ablations that toggle each core idea of their algorithm?
- Other questions are raised under Weaknesses.

---

### Official Review · Reviewer_xJGJ · 2025-11-03

**Soundness:** 1
**Presentation:** 2
**Contribution:** 1
**Rating:** 2
**Confidence:** 4

**Summary:**

This paper proposes an approach of combining Evolutionary Algorithms and Reinforcement Learning, called EPO. EPO maintains a population of policies, consisting of a master policy that learns in an off-policy manner and other policies are optimized via EA. The policies in the population share the policy and critic networks, and each policy owns its unique gene embedding. The proposed method is empirically evaluated in 8 tasks, including dexterous manipulation, legged locomotion, and classic control.

**Strengths:**

- The presentation of the proposed method is almost clear.

**Weaknesses:**

- The motivation of this work does not make sense to me. I understand the value of combining EA and RL. But the authors motivate this point by mentioning that on-policy RL does not scale well with larger batch sizes. However, off-policy RL is not discussed in this logic chain.
- Combining EA and RL has been well studied and a comprehensive survey is provided in [7], as well as other related topics like RL with parallel/population environments. However, important related literature is largely ignored in this paper.
    - RL with vectorized environments
        - [1] Simplifying Deep Temporal Difference Learning. https://arxiv.org/abs/2407.04811
        - [2] FastTD3: Simple, Fast, and Capable Reinforcement Learning for Humanoid Control. h[ttps://arxiv.org/abs/2505.22642](https://arxiv.org/abs/2505.22642)
    - Evolutionary RL
        - [3] Policy optimization by genetic distillation. ICLR 2018
        - [4] EvoRainbow: Combining Improvements in Evolutionary Reinforcement Learning for Policy Search. ICML 2024.
        - [5] Proximal distilled evolutionary reinforcement learning. AAAI 2020.
        - [6] ERL-Re$^2$: Efficient Evolutionary Reinforcement Learning with Shared State Representation and Individual Policy Representation. ICLR 2023
        - [7] Bridging Evolutionary Algorithms and Reinforcement Learning: A Comprehensive Survey on Hybrid Algorithms. *IEEE Transactions on Evolutionary Computation (TEC)*
- Line 156 is problematic. Infinite horizon with a discount factor of 1 leads to numerical divergence
- Concerns on the experiments:
    - For performance comparison, important baseline methods that use evolutionary RL or population-based RL, such as ERL-Re2 [6], PDERL [5], EvoRainbow [4], FastTD3 [2], should be included as necessary baselines. The authors claim “EPO outperforms state-of-the-art baselines in sample efficiency, asymptotic performance, and scalability”. From my perspective, this is an overclaim with no convincing support.
    - The “ablation study” provided in Figure 4 is not an ablation. The ablation should be comparing EPO with its variants like EPO - off-policy training, EPO - mutation, EPO - crossover, etc.
    - The scaling provided in Figure 4 does not make sense to me, because more samples for used when increasing the number of agents or environments and the convergence performance of the curves is not presented. Thus, I cannot rule out the possibility that “32 agents” converges the same/learns better than “64 agents” when using the same number of interaction samples.
    - Moreover, why is “128 agents“ only shown in Figure 4(b) but missing in Figure 4(a)?
    - One very important point is about diversity. The authors mentioned multiple times that the recipe of EPO (or generally evolutionary RL) is the diversity provided by the population. However, I did not see any in-depth analysis on the differences or diversity among policies/genes/the trajectories generated by different policies. This is very important to the evaluation of the proposed method.

**Questions:**

1. For crossover operation, the authors mentioned “we use a simple averaging method for efficiency”. This averaging strategy does not fit biological intuition. An experimental comparison on different choices of crossover operators should be included.
2. How is the mutation noise decided and how do different choices influence the results?
3. The authors mentioned “policies require longer training before they exhibit meaningful differences in performance … we execute selection, crossover, and mutation only when the agents’ performance differences become sufficiently significant” near Line 249. Since genetic operations are not applied until the differences are significant enough, where do the differences come from? This is somewhat counter-intuitive because it is waiting for sufficient diversity/differences to apply the genetic algorithm which aims to enhance diversity.
4. Can the authors provide the analysis on the differences or diversity among policies/genes/the trajectories generated by different policies? In my opinion, this should be a very critical part of the experiments.

---

### Meta-Review · Area_Chair_RsDe · 2026-01-03

**Summary:**

Here is a summary of the reviewers' concerns:
- Limited novelty: Combining EA and RL has been well studied and a comprehensive survey is provided in [7].
- Limited experiments: missing comparison with some baselines such as ERL-Re2 [6], PDERL [5], EvoRainbow [4], FastTD3 [2].
- Lack theoretical justification: Section 4.2's off-policy loss needs an algorithmic or theoretical justification that it doesn't break any assumptions of the underlying algorithms and would preserve its convergence properties.
- Missing ablation study to identify the source of improvement.
- Lack deep analysis on the diversity among genes/trajectories generated by different policies, which is important to the evaluation of the proposed method.
- Insufficient scaling evidence: §5.4’s scaling test (up to 49 k envs) lacks baseline comparisons and deeper analysis.
- The evolutionary trigger using range/median is heuristic; a sensitivity scan over γ and a comparison with fixed-interval trigger or trend-based trigger are needed, especially in sparse reward environments.
- Missing technical details: recommend consolidating list of elite count, crossover method, mutation noise, latent dimension & injection location.
- Lack motivation.

**Reviewer Concerns:**

The authors did not make a rebuttal. All reviewers' concerns remain.

**Reviewer Scores:**

All reviewers are likely to keep their scores since there is no rebuttal.

---

### Decision · Program_Chairs · 2026-01-26

Reject